# Cross-Platform Transcriptomic Data Integration, Profiling, and Mining in *Vibrio cholerae*

Zi-Xin Qin,[a] Guo-Zhong Chen,[a] Qian-Qian Yang,[a] Ying-Jian Wu,[b] Chu-Qing Sun,[b] Xiao-Man Yang,[a] Mei Luo,[a] Chun-Rong Yi,[a] Jun Zhu,[a] Wei-Hua Chen,[b] Zhi Liu[a]

[a]Department of Biotechnology, College of Life Science and Technology, Huazhong University of Science and Technology, Wuhan, Hubei, China

[b]Department of Bioinformatics and Systems Biology, Huazhong University of Science and Technology College of Life Sciences and Technology, Wuhan, Hubei, China

Zi-Xin Qin and Guo-Zhong Chen contributed equally to this work. Author order was determined by the corresponding authors after negotiation.

**ABSTRACT** A large number of transcriptome studies generate important data and information for the study of pathogenic mechanisms of pathogens, including *Vibrio cholerae*. *V. cholerae* transcriptome data include RNA-seq and microarray: microarray data mainly include clinical human and environmental samples, and RNA-seq data mainly focus on laboratory processing conditions, including different stresses and experimental animals *in vivo*. In this study, we integrated the data sets of both platforms using Rank-in and the Limma R package normalized Between Arrays function, achieving the first cross-platform transcriptome data integration of *V. cholerae*. By integrating the entire transcriptome data, we obtained the profiles of the most active or silent genes. By transferring the integrated expression profiles into the weighted correlation network analysis (WGCNA) pipeline, we identified the important functional modules of *V. cholerae in vitro* stress treatment, gene manipulation, and *in vitro* culture as DNA transposon, chemotaxis and signaling, signal transduction, and secondary metabolic pathways, respectively. The analysis of functional module hub genes revealed the uniqueness of clinical human samples; however, under specific expression patterning, the Δ*hns*, Δ*oxyR1* strains, and tobramycin treatment group showed high expression profile similarity with human samples. By constructing a protein-protein interaction (PPI) interaction network, we discovered several unreported novel protein interactions within transposon functional modules.

**IMPORTANCE** We used two techniques to integrate RNA-seq data for laboratory studies with clinical microarray data for the first time. The interactions between *V. cholerae* genes were obtained from a global perspective, as well as comparing the similarity between clinical human samples and the current experimental conditions, and uncovering the functional modules that play a major role under different conditions. We believe that this data integration can provide us with some insight and basis for elucidating the pathogenesis and clinical control of *V. cholerae*.

**KEYWORDS** *Vibrio cholerae*, computational biology, WGCNA

Address correspondence to Zhi Liu, zhiliu@hust.edu.cn, or Wei-Hua Chen, weihuachen@hust.edu.cn.

The authors declare no conflict of interest.

The human intestinal pathogen *Vibrio cholerae* is still endemic in developing countries, such as Haiti and Zimbabwe, where adequate clean municipal water supply systems are scarce (1). With the development of high-throughput sequencing techniques, tracking the dynamics of these pathogens in infections has become possible, which contributes to pathogenesis mechanism research and lays a concrete foundation for pathogen prevention and control. Clinical human microarray data published in 2002 revealed high expression of nutritional uptake and motility genes and low expression of chemotaxis genes during human infection (2). In 2005, clinical human microarray data further verified higher expression of virulence genes, such as the toxin

co-regulated pilus gene *tcp*, and hemolysin genes during early infection (3). *In vivo* RNA-seq data of rabbits and mice published in 2011 investigated the importance of cholera toxins (CT) and toxin-coregulated pilus (TCP) during *V. cholerae* colonization in animal models, as well as high expression of nutrient competition pathways, and discovered expression differences between animal models (4). To date, the most important clinical data exist in microarray, but current laboratory studies mainly use RNA-seq.

Multiomics big data analysis of *V. cholerae* has now been applied in RNA-seq, transposon insertion sequencing (Tn-seq), and chromatin immunoprecipitation sequencing (ChIP-seq). DuPai et al. (5) performed weighted correlation network analysis (WGCNA) by integrating these data, then mapping the functional set of *V. cholerae* with an interplay network and incorporating new unknown functional genes. WGCNA can uncover similar gene expression patterns across different samples and form highly correlated co-expressed gene modules, correlating the modules with each other and with external sample traits to find candidate biomarkers (6). From these, combining Tn-seq and ChIP-seq, a more detailed virulence gene interactions network was produced, and many virulence candidate genes were predicted. *In vitro* experiments on pathogenic bacteria are expected to simulate infection in humans as closely as possible. Unfortunately, transcriptome studies of *V. cholerae* in clinical patients mainly use the microarray, while RNA-seq is mainly used in laboratory samples. And the principles of these two techniques are different, resulting in an inability to compare laboratory data with data from precious clinical samples.

Tang et al. (7) developed the Rank-in algorithm by converting raw expression to a relative ranking in each profile and then weighting it according to the overall expression intensity distribution in the combined data set. By minimizing analytical differences between array and RNA-seq for cross-platform data integration, they achieved the greatest accuracy, precision, and recall compared to a previous representative procedure that eliminated non-biological effects which was validated in a colon cancer data set. Castillo et al. (8) obtained RNA-seq and microarray expression values separately by a standard process, then combined and homogenized them using the Limma R package normalizedBetweenArrays function, which was finally validated in a breast cancer data set.

Here, we have used these two techniques to integrate *V. cholerae* RNA-seq data for laboratory studies with clinical microarray data for the first time. We believe that this data integration can provide us with some insights and basis for elucidating the pathogenesis and clinical control of *V. cholerae*.

## RESULTS

**Analysis procedures.** For initial raw data collection, we downloaded and curated all *V. cholerae* RNA sequencing data manually deposited in NCBI's Sequence Read Archive (SRA) (9) and literature supplemental materials from PubMed, as well as *V. cholerae* gene expression from the Gene Expression Omnibus (GEO) database. All RNA-seq samples were mapped to a recently inferred *V. cholerae* transcriptome derived from the N16961 reference genome by Kallisto (10). We treated the two channels of the microarray data as independent experiments, then used an in-house R script to perform the normalization. We integrated data from two different platforms via Rank-in and the Limma R package normalizedBetweenArrays function. Next, we applied WGCNA (https://github.com/ShawnWx2019/WGCNA-shinyApp) pipeline to analyze the entire gene expression table and generate a co-expression network. The literature mining and manual curation workflow is shown in Fig. 1.

**Cross-platform data integration.** Through boxplot analysis, we found that there was a heterogeneous distribution of gene expression between the different platforms and therefore they could not be used for comparison (Fig. 2A). For example, RNA-seq is the gene count value, ranging from 0 to 20,000, while microarray is the probe fluorescence value, ranging from 0 to 14. Even within the platform, there is heterogeneous distribution of gene expression between different projects and therefore they cannot be directly used for comparison. Through the R t-SNE function, we also found that there was a batch effect in the expression profile between different batches of items from RNA-seq and microarray, i.e., self-aggregation of identical items (Fig. 2B). We normalized these data using the Rank-in algorithm (7), and the results showed a shift from

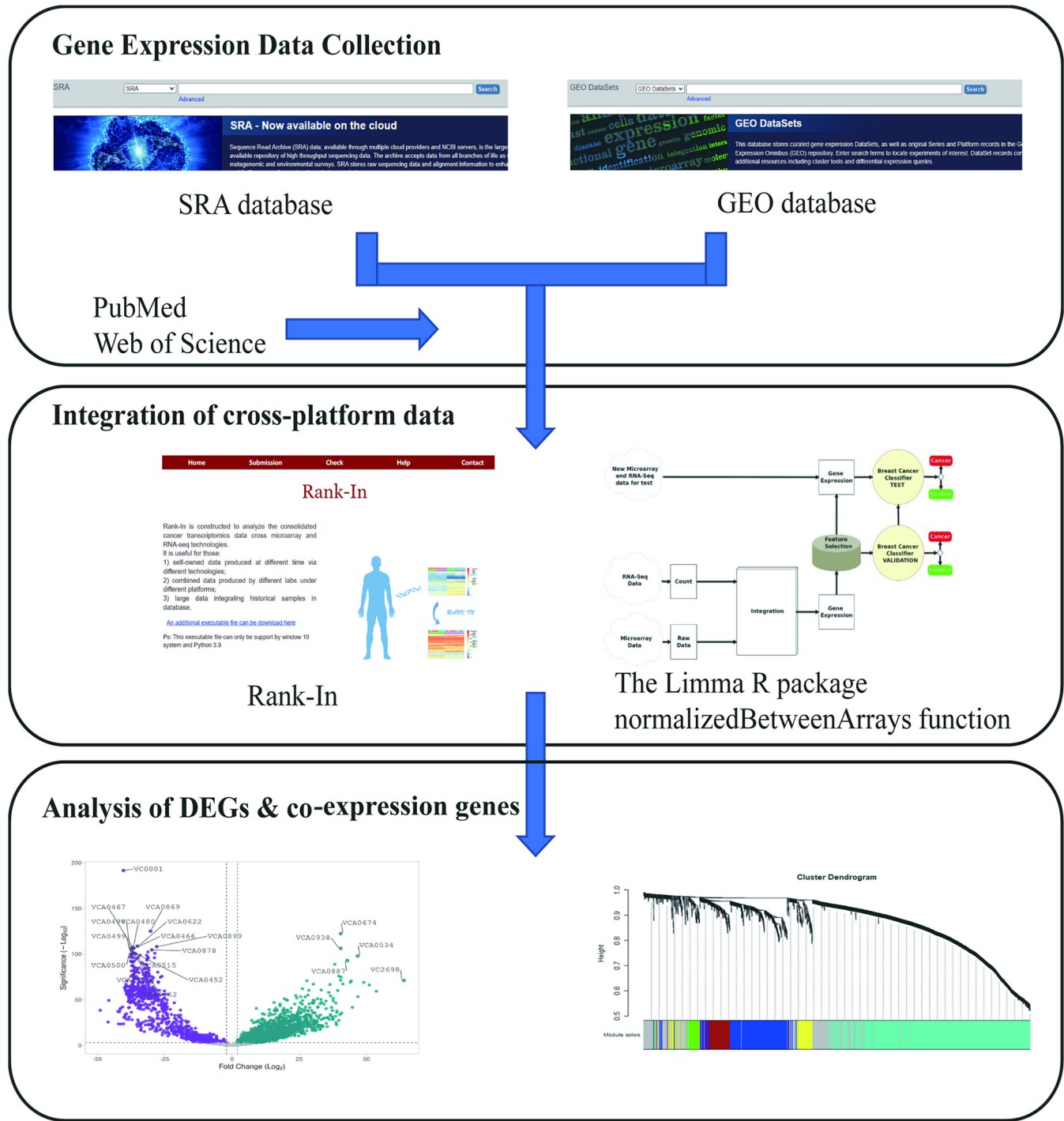

**FIG 1** Workflow of the integrated transcriptome. The *V. cholerae* transcriptome data were downloaded through the SRA and GEO databases and supplemented through PubMed and Web of Science. Raw data were processed to obtain expression matrices that were entered into Rank-in or the Limma R package normalizedBetweenArrays function for integration. Integrated expression matrices were used to construct co-expression networks by WGCNA for subsequent analysis.

self-aggregation of same-item (same-colored) samples with batch effects to sample dispersion across groups after mitigation of batch effects (Fig. 2C). The t-SNE down-scaled samples appeared to be mixed between samples from different platforms (different-colored) (Fig. 2D). These results indicate that we have integrated data originating from different projects between and within the RNA-seq and microarray platforms through the Rank-in algorithm, which can be used for further analysis.

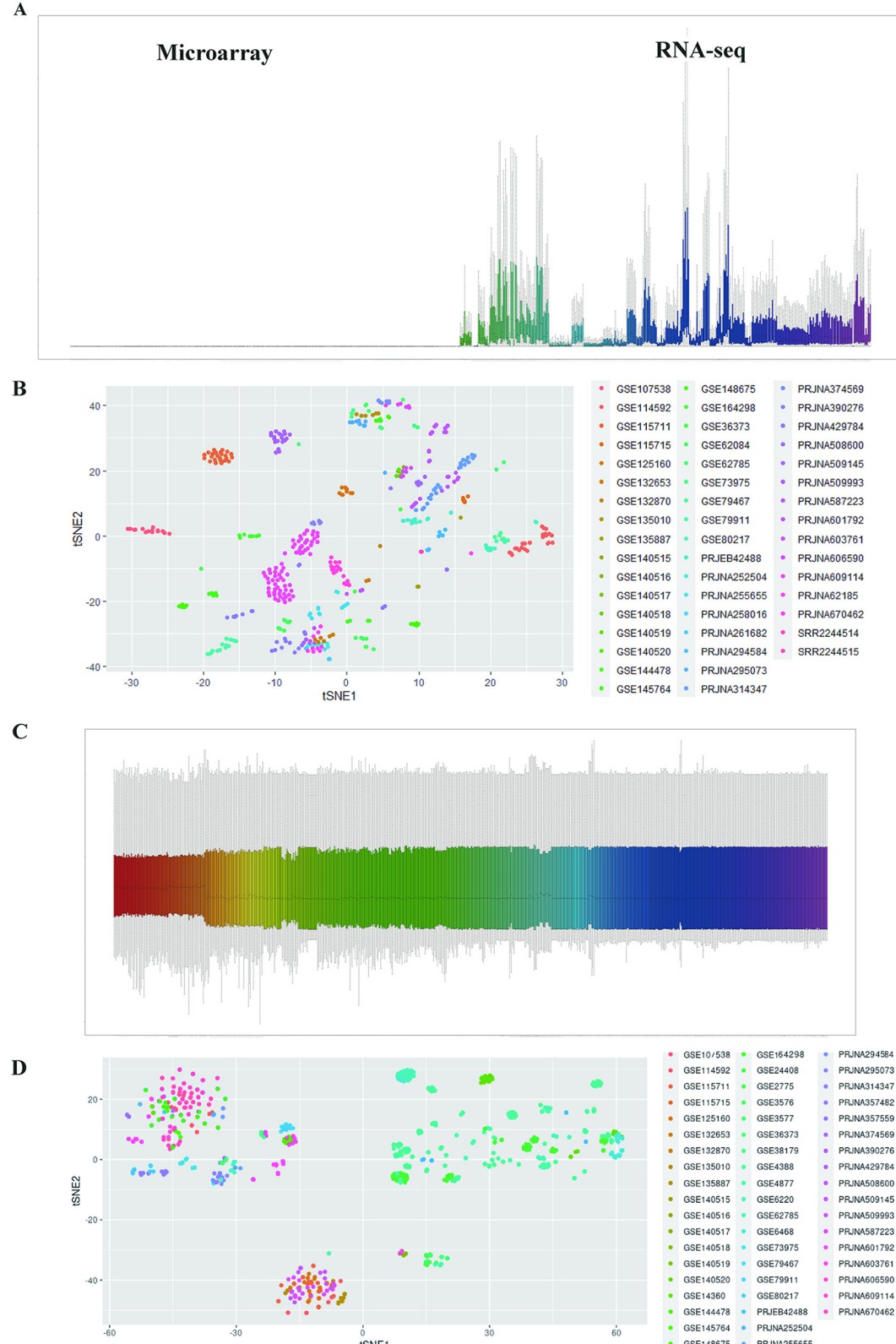

**FIG 2** Integrating data for cross-platform comparability. (A) Transcripts of samples before integration, with RNA-seq samples on the right (count in horizontal coordinates) and microarray samples on the left (fluorescence values in horizontal coordinates). (B) Spatial distribution of unintegrated whole samples after R t-SNE downscaling. (C) Transcripts of all samples after integration (horizontal coordinates are adjusted gene expression values). (D) Spatial distribution of all samples after integration by R t-SNE dimensionality reduction.

**Cross-platform transcriptomic data profiling.** In total, we obtained transcript profiles for 59 projects and 873 samples of *V. cholerae* from 2002 to June 2021, which contained expression data for 3,885 genes. Some genes were not included due to absence or low expression in the microarray cDNA template. We first examined the profile of the *V. cholerae* gene transcriptome in the context of the entire complex sample. The overall transcripts identified ~2,000 differentially expressed genes (DEGs) in Fig. 3A. DEGs are calculated by a nonparametric approach (Wilcoxon rank-sum test). We used an false discovery rate (FDR) threshold of 0.05 for multiple testing. We screened for ~2,000 DEGs occupying all *V. cholerae* genes (~3,894). It is implied that most of the *V. cholerae* genes are active in different environments.

The DEGs were functionally enriched by the Gene Ontology (GO) and Kyoto Encyclopedia of Genes and Genomes (KEGG) databases. The GO database (Fig. 3B) was found to be clustered mainly on ATP-binding, cytoplasmic, and ribosomal structures, which represent energy metabolism, intracellular physiological activities, and transcription in *V. cholerae*, respectively. KEGG functional enrichment (Fig. 3C) showed that the metabolic activity and secondary metabolite synthesis pathways were the most represented functional pathways, suggesting that the metabolic profile of *V. cholerae* also changes in different environments. Both GO and KEGG point to nutrient uptake and utilization and energy metabolism by *V. cholerae*, indicating their importance in its entire life history.

The top five genes most significantly upregulated at the overall transcriptome level were *vc2698*, *vca1031*, *vc0910*, *vc0911*, and *vc2699*. The gene *vc2698* (*aspA*), an aspartate ammonia-lyase, was identified in a previous study (11) as being responsible for the reversible conversion of aspartate to ammonia and fumarate, thereby regulating cell-wall precursor synthesis to alter cell morphology in *V. cholerae*. *vca1031* has been annotated as a methyl-accepting chemotaxis protein commonly associated with nutritional competition (12), but its function has not been investigated. *vc0910* (*treB*) was identified as phosphotransferase system (PTS) trehalose transporter subunit IIBC and *vc0911* (*treC*) was identified as $\alpha,\alpha$-phosphotrehalase; both of these belong to the PTS system, which is essential for the colonization of *V. cholerae*, especially in the host (13). *vc2699* is named as an anaerobic C4-dicarboxylate transporter in *Escherichia coli*, a component of the bacterial two-component system which has previously been investigated in *E. coli* (14). The top five activated genes are all related to nutrient transformation and utilization, suggesting an important role of nutrient competition in *V. cholerae* growth and colonization.

The top five genes most significantly downregulated at the overall transcriptome level were *vc0488*, *vc0063*, *vc1704*, *vc2642*, and *vc0386*. *vc0488* is annotated as TRAP transporter substrate-binding protein, and its function has not been reported. *vc0063* is annotated as thiazole biosynthesis adenylyltransferase ThiF, which belongs to thiamine metabolism in the KEGG metabolic pathway, but its function in *V. cholerae* is also not reported. *vc1704* is annotated as 5-methyltetrahydropteroyltriglutamate-homocysteine *S*-methyltransferase MetE, which is regulated by the *metR* gene in *E. coli* and activates the methionine synthesis pathway together with *metH*. *vc2642* is annotated as argininosuccinate synthase and is involved in the aspartate and citrulline metabolic cycle. *vc0386*, annotated as phosphoadenylyl-sulfate reductase, appears to be involved in the regulation of bacterial redox resistance, but its function in *V. cholerae* has also not yet been explored. Although these genes have not been studied in *V. cholerae*, the annotation information suggests that most of them still belong to nutrient metabolism, especially amino acid metabolism.

The internal reference genes are very important for transcriptional-level studies, and a good internal reference gene needs to meet a certain amount of expression and stable expression. The common internal reference genes selected for *V. cholerae* transcription were *ompT* (*vc0862*) (15), *rpoA* (*vc2571*) (16, 17), *gyrB* (*vc0015*) (18, 19), *thyA* (*vc0675*) (20), and *recA* (*vc0543*) (21). Under all conditions, we found that the expression levels of these internal reference genes were not stable under specific conditions

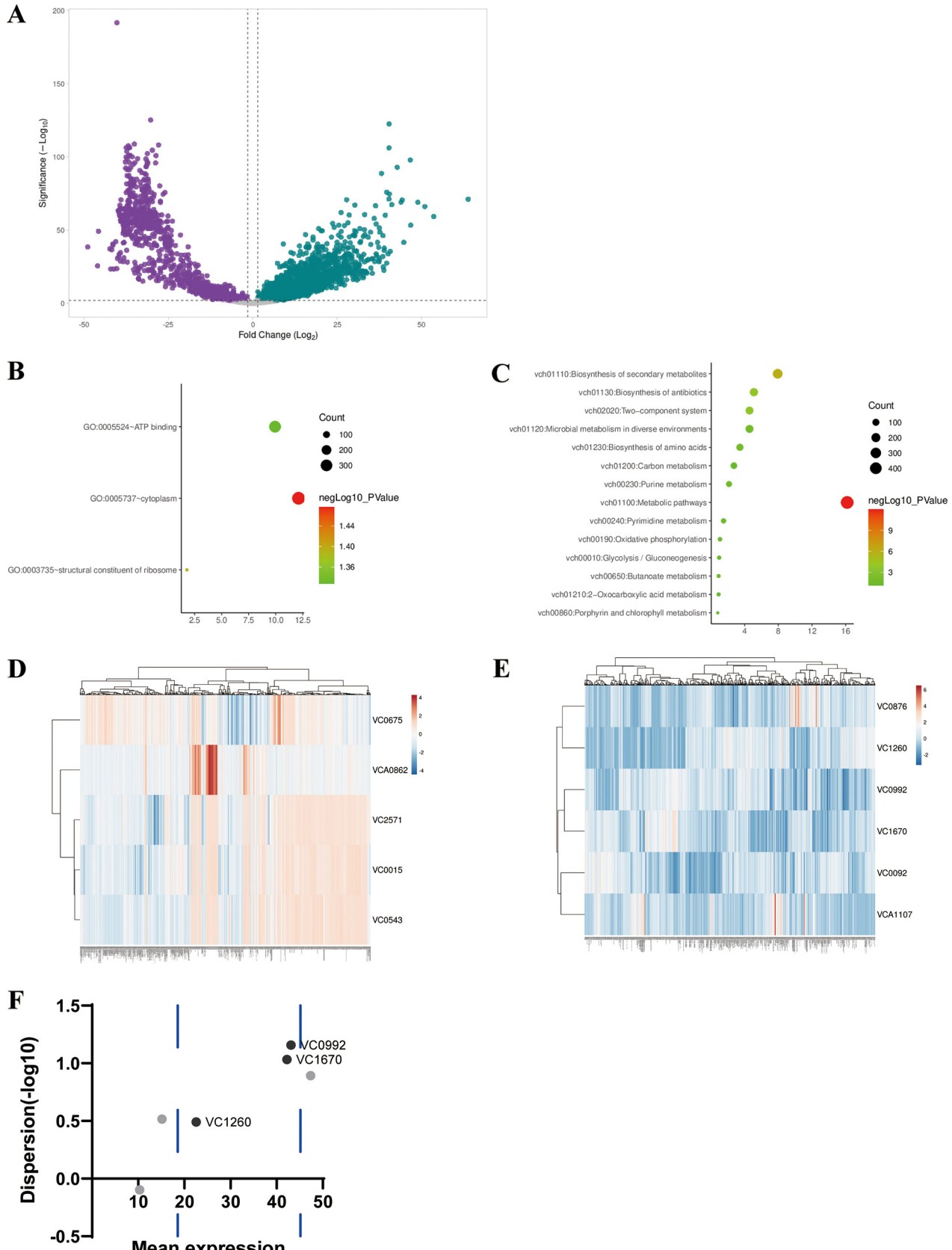

**FIG 3** Integral differentially expressed genes reveal dynamic changes in functional pathways in *V. cholerae*. (A) Volcano plot of differentially expressed genes identified from the entire transcriptome after integration, with differential expression at $P < 0.05$ and log$_2$-fold change greater

(Fig. 3D). These genes exhibit varied expression under specific experimental conditions and therefore need to be selected for suitable experimental conditions. By selecting the genes with the smallest fold-change in unDEGs, we obtained several candidate genes (Fig. 3E). We also investigated the average expression level and dispersion of all genes at the overall transcriptional level (Fig. S1), screened genes with moderate expression by quartiles, and sorted them by dispersion to obtain candidate genes with low dispersion and suitable expression. Among these genes, *vc0876*, *vc0092*, and *vca1107* were excluded because their expression mean deviated from the intermediate state, and *vc1260* was eliminated due to high dispersion. We found that *vc0992* and *vc1670* performed well under both criteria (Fig. 3F). Gene *vc0992* is annotated as a cation-proton antiporter. Gene *vc1670* is annotated as cardiolipin synthase (CLS), which is responsible for the formation of bacterial cell membranes.

**WGCNA reveals new functional clusters in *V. cholerae*.** For tumor transcriptional information mining, there is often a clear distinction between sample traits such as tumor tissue and normal tissue. However, studies of pathogenic bacteria at the transcriptional level involve a variety of treatments. Therefore, samples with similar expression patterns are categorized based on their integrated overall sample expression profiles to facilitate study of the association of gene expression with sample traits. The different samples showed clustering in the R t-SNE reduced-dimensional bipartite graph, with four groups obtained by hierarchical clustering (Table S5) and the clustering displayed in the reduced-dimensional t-SNE plot (Fig. 4A). Group 1 is dominated by *in vitro* culture samples, at about 75%, with the remaining portion consisting of *in vitro* stress (9%), genetic manipulation (6%), and *in vivo* samples (9%). Group 2 is dominated by *in vitro* culture (35%) and *in vitro* stress samples (46%), with the remaining portion consisting of *in vivo* (11%) and genetic manipulation samples (8%). Group 3 is dominated by genetic manipulation samples, at about 66%, with the remaining portion consisting of *in vitro* culture (23%), *in vitro* stress (10%), and *in vivo* samples (1%). Group 4 contained no *in vivo* samples and consisted of *in vitro* culture (63%), genetic manipulation (31%), and *in vitro* stress samples (6%). The co-expressed gene network in *V. cholerae* transcripts was revealed by the WGCNA pipeline, which distributed the filtered genes into 6 modules (Fig. 4B); the correlation between different modules was calculated, and the different modules were more independent from each other except for the yellow and green modules (Fig. S2). Subsequently, to explore the reasons for the clustering of different groups, the gene function modules were correlated with groups (Fig. 4C): group 1 was positively correlated with the gray module, group 2 with the turquoise module, group 3 with the blue module, and group 4 with the brown module. Genes which were not co-expressed with genes in other modules are grouped into gray modules. Therefore, when similar experimental treatments are available, genes within modules that are positively correlated with the group to which the treatment belongs may preferentially be the target of detection.

The different transcriptome samples of *V. cholerae* can be artificially divided into the main categories of stress conditions, *in vitro* culture, *in vivo* host, and gene manipulation. Although these samples were dispersed within groups by hierarchical clustering, we wanted to investigate whether treatment conditions within the same group correlated with gene expression patterns. After removing the different sample types from each group, we calculated the correlation between modules and groups (Fig. 4D). We found that removing the *in vitro* culture samples significantly reduced the correlation between the second group and the turquoise module and between the fourth group and the brown module. Removing the stress samples significantly reduced the correlation

**FIG 3** Legend (Continued)
than a 2-bit threshold. (Horizontal coordinates are –log*P* values, vertical coordinates are $\log_2$-fold change values) (B) GO and (C) KEGG enrichment of differentially expressed genes. (D) Heatmap of the expression of the five commonly used internal reference genes in the whole transcriptome. (E) Heatmap of expression of selected candidate internal reference genes in the whole transcriptome. (F) Expression mean values (vertical coordinate) and dispersions (horizontal coordinate) of selected genes; dashed blue lines indicate quartiles of gene expression means. Genes outside the suitable expression mean interval are shown in gray.

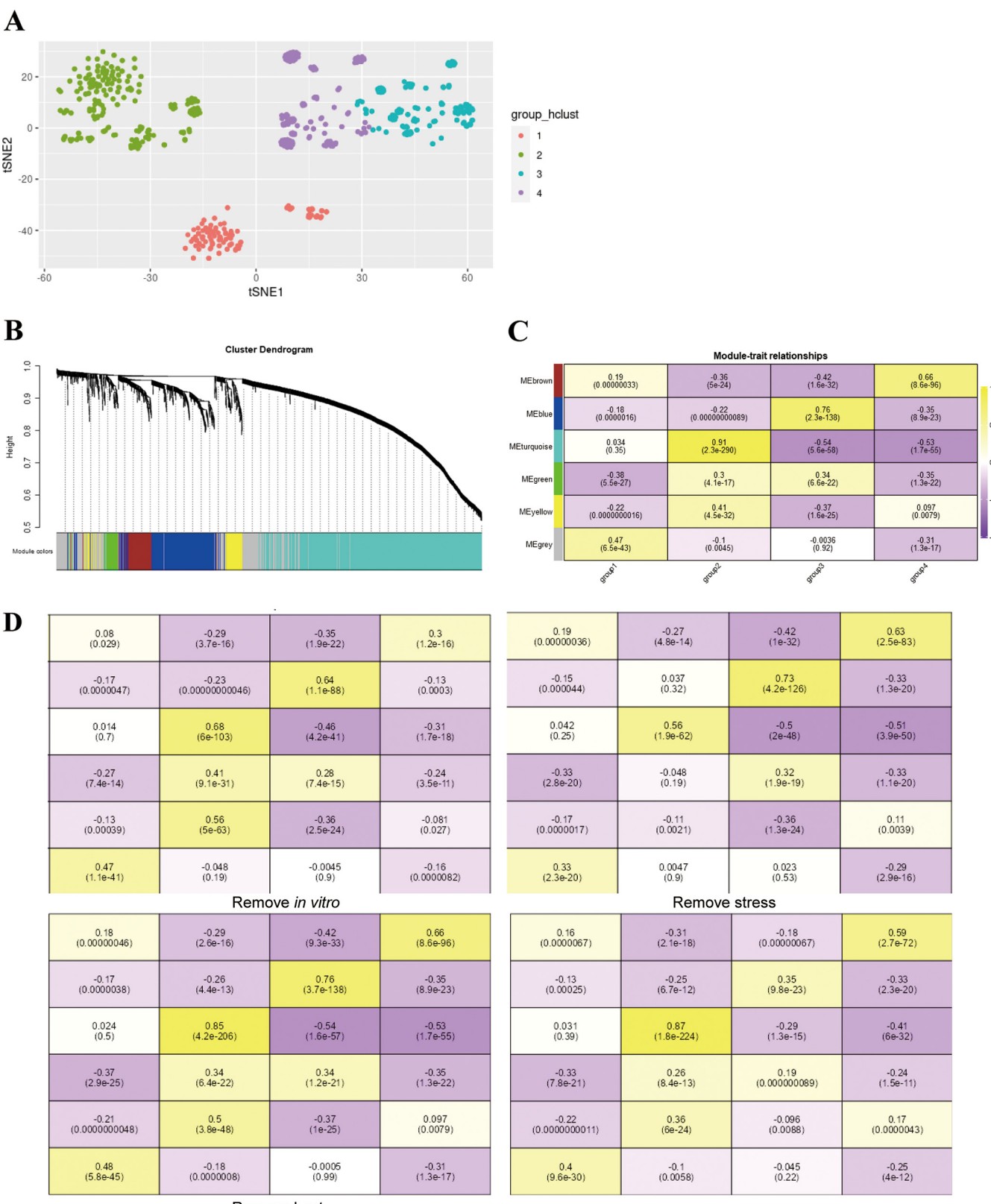

**FIG 4** WGCNA reveals functional module and sample clustering correlations. (A) Distribution of the four groups of samples after hierarchical clustering on a t-SNE plot, with four colors representing the four groups (red, group 1; green, group 2; turquoise, group 3; purple, group 4). (B) Clustering dendrogram of different genes identified by WGCNA assigned to different modules. (C) Correlation between modules and groups. Upper values in the boxes are correlation coefficients, lower values are *P* values. (D) Correlation between modules and groups after removing samples of different categories in different groups.

between the second group and the turquoise module. Removing the gene manipulation significantly reduced the correlation between the third group and the blue module. Interestingly, the removal of samples from the host did not seem to have a significant effect on the correlations of the groups. The removal analysis suggested that under *in vitro* culture and stress conditions, gene function within the turquoise module is more important than that in other modules. Under genetic manipulation conditions, it is possible that the correlation between genes within the blue module and target genes is weak, as this correlation is commonly found within genetic manipulation samples.

**WGCNA hub genes analysis elucidated the main functions of modules.** Co-expression analysis of the genes within the module yielded correlated co-expressed genes. Identification of the hub genes within a module helps us understand the function represented by the module's gene expression (22). The hub genes of three modules were obtained by screening with |gene significance (GS)| > 0.2 and |module membership (MM)| > 0.8 and were analyzed for functional enrichment. The turquoise module (Fig. 5A) was mainly enriched for functional pathways, including transposases, DNA-mediated transpositions, acetyltransferases, and other DNA-level changes and membrane essential components. The blue module (Fig. 5B and C) was mainly enriched for functional pathways, including chemotaxis, signal transduction, and metabolism-related pathways. The brown module (Fig. 5D and E) was enriched for functional pathways, including cytoplasmic, signal transduction, metabolism-related, and metabolic pathways of bacteria in response to different environments.

**Using hub gene transcript-level analysis to analyze the association between laboratory and clinical samples.** Clinical human samples are undoubtedly the most important part of pathogenic bacterial transcriptome data, and it is interesting to note that human samples remain in a unique position in the integrated data set, with little similarity to other samples. This also suggests that current experimental conditions can only partially mimic the human gut environment. Current animal and *in vitro* models for studying *V. cholerae* pathogenesis do not perfectly mimic clinical infections. Therefore, we wanted to discover which treatment outcomes are closer to *in vivo* clinical condition through omic data mining. Figure 4A shows that the human data are assigned into group 1, implying that the expression patterns of group 1 samples are closer to those of human samples. Furthermore, we used principal-component analysis (PCA) to determine the similarity between the group 1 samples and human samples, and found that the human samples were more isolated from other samples. Because hub genes represent the most dominant gene change in an expression pattern, we considered whether they could be used as a biomarker to observe experimental conditions which resemble those of the host in this expression pattern. The transcriptional profiles of the corresponding hub genes within the host and the samples within each group were extracted and subjected to PCA, and we found that the other samples in the turquoise module expression pattern of group 2 were still significantly different from the clinical samples (Fig. 6A). However, the expression profiles of the $\Delta oxyR1$ and $\Delta hns$ strains in the blue module expression pattern of group 3 were most similar to the clinical samples (Fig. 6B), and the brown module expression pattern of group 4 tobramycin treatment seemed to be closer to the clinical samples in the brown module expression pattern of group 4 (Fig. 6C).

To validate this conclusion, we used newly sourced microarray samples of *V. cholerae* infecting humans (2), finding that 50% of the genes (5/10) in the $\Delta hns$ strains were also present in the new human samples (Fig. 6D) and ~35% of the genes (20/56) in the tobramycin samples were also in human samples (Fig. 6E). This demonstrates the reliability of our analytical approach, although many biological experiments will eventually be needed to verify the roles of these genes in the colonization of *V. cholerae*.

**PPI interaction networks between hub genes suggest novel interactions.** Co-expressed genes often have interactions or regulatory relationships with each other, so we constructed protein-protein interaction (PPI) networks from hub genes identified in WGCNA using STRING (23), a current collection of known and predicted direct physical

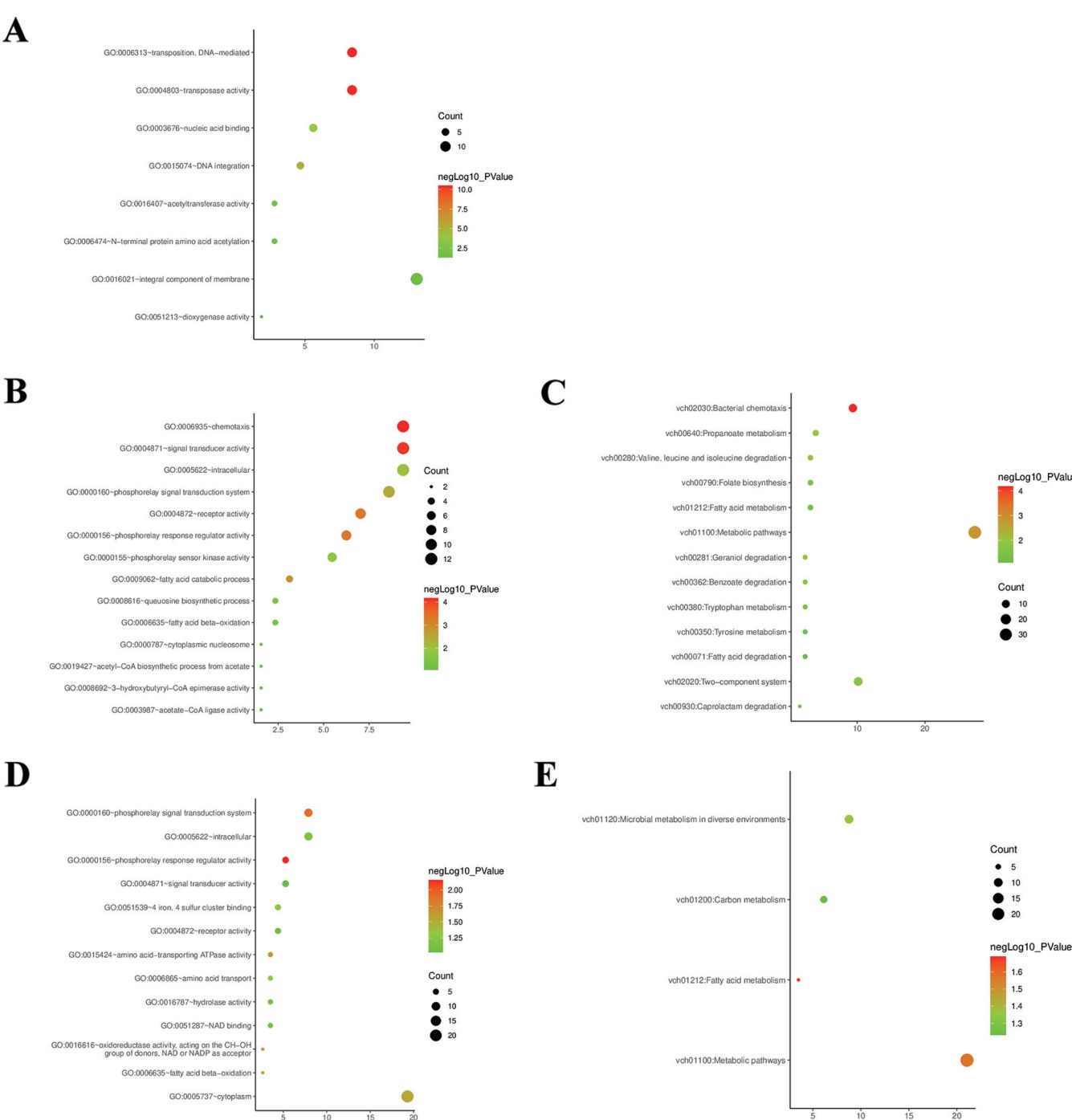

**FIG 5** Hub gene suggests the main functions of the module. (A) GO functional enrichment of hub genes within turquoise module. (B) GO and (C) KEGG enrichment results from hub genes within the blue module. (D) GO and (E) KEGG enrichment results from hub genes within the brown module.

binding and indirect functionally related interactions between proteins/genes. The PPI networks constructed for the blue and brown modules of hub genes exhibit tight interactions where functional enrichment is consistent with the preceding GO/KEGG (Fig. S3B and C). Interestingly, relatively few interactions were identified in the PPI network of the hub genes from the turquoise module (Fig. S3A). We performed PPI network mapping of hub genes within the three modules by gene co-expression relationships obtained from WGCNA, with a weight of top 10% as the threshold (Fig. 7A, C, and E). The PPI interactions between the two different sources were also compared separately. We found that the proportion of PPIs shared by both within the blue module was the

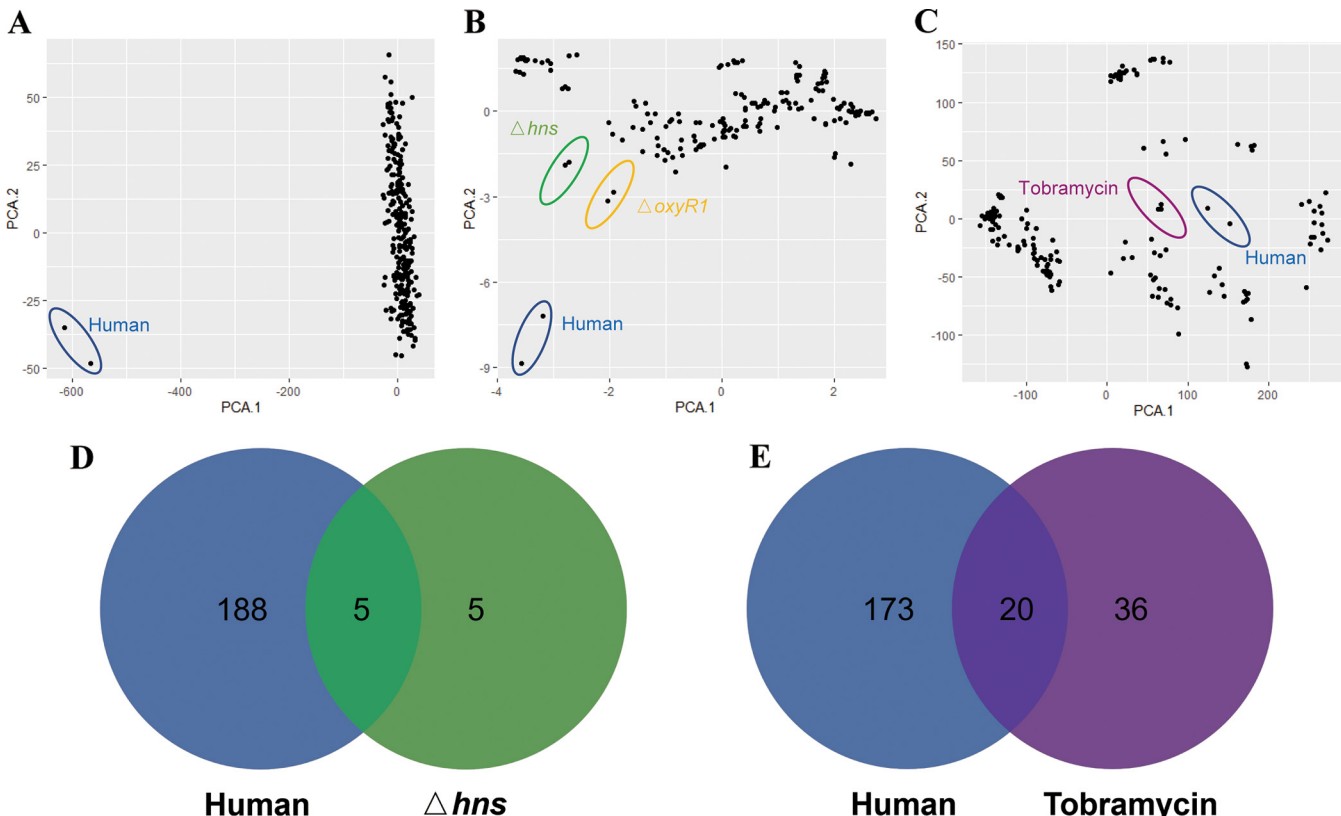

**FIG 6** Hub gene can be used as a conditional biomarker. (A) Principal-component analysis plot of the hub gene expression profiles of group 2 samples versus clinical samples. Blue circles are clinical samples. (B) Principal-component analysis of the hub gene expression profiles of group 3 samples versus clinical samples. Blue circles are clinical samples, green circles are Δ*hns* strains, and yellow circles are Δ*oxyR1* strains. (C) Principal-component analysis of the hub gene expression profiles of group 3 samples and clinical samples. Blue circles are clinical samples, purple circles are tobramycin-treated samples. (D) The number of DEGs shared between new human samples and Δ*hns* samples, the new human samples on the left and the Δ*hns* samples on the right. The DEGs for the Δ*hns* samples must also be hub genes. (E) The number of DEGs shared between new human samples and tobramycin samples, the new human samples on the left and the tobramycin samples on the right. The DEGs for the tobramycin samples must also be hub genes.

highest, accounting for 12.6% of the WGCNA source and 29.2% of the STRING source, respectively (Fig. 7D). The brown module was the next highest, with 8.8% of the shared PPIs for the WGCNA source and 9.8% for the STRING source, respectively (Fig. 7F). The turquoise module had the least shared PPIs, accounting for 0.26% of the WGCNA source and 10.5% of the STRING source, respectively (Fig. 7B). The variability of this co-expression relationship within the turquoise module with the data from STRING may suggest the existence of many novel protein interactions within the turquoise module, pending validation in subsequent experiments.

## DISCUSSION

We have successfully integrated transcriptomic data of *V. cholerae* from both platforms through Rank-in and thus constructed a gene co-expression network for *V. cholerae* using WGCNA. This cross-platform data integration allows us to bring a more diverse perspective to the study of the *V. cholerae* transcriptome in response to different environmental changes, while also incorporating the most important clinical human samples in pathogenic bacteria research. Our integrated data set includes the shortcomings of currently used transcriptional internal reference genes under certain specific experimental conditions in a holistic perspective, while providing several genes that are relatively consistently expressed under all experimental conditions for reference purposes. Also, after removing batch effects and inter-platform heterogeneity, we identified auto-clustering among different samples, analyzed them, and constructed gene co-expression networks, from which functional modules and hub genes with high relevance to the samples were screened and identified. We also constructed

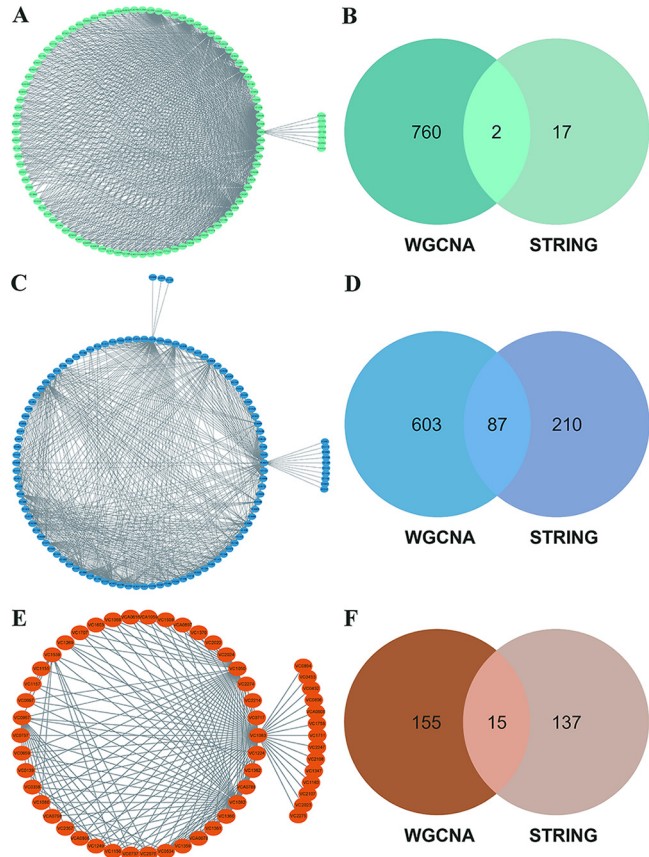

**FIG 7** Protein-protein interaction (PPI) interaction networks between hub genes suggest novel interactions. (A) PPI network diagram of hub genes within the turquoise module. (B) Number of PPIs from hub genes in turquoise module shared between the two sources, with the WGCNA source on the left and the STRING source on the right. (C) PPI network diagram of hub genes within the blue module. (D) Number of PPIs from hub genes in blue module shared between the two sources, with the WGCNA source on the left and the STRING source on the right. (E) PPI network diagram of hub genes within the brown module. (F) Number of PPIs from hub genes in brown module shared between the two sources, with the WGCNA source on the left and the STRING source on the right.

PPI networks to predict the possible existence of unstudied gene interactions. Interestingly, similar results can be obtained by integrating the data through the Limma R package normalizedBetweenArrays function (Fig. S6A to E).

Although the logic for integrating data differs between the two pipelines, the main highly correlated modules are quite similar, such as group 2 with the turquoise module and group 3 with the blue module (Fig. S6F). The hub genes within the modules are shown as a Sankey diagram, where the turquoise and blue modules are nearly identical to the average gene in both merging methods, and the brown module is not as similar to the green module (Fig. S6G).

The functions enriched by hub genes within the turquoise module mainly focus on DNA-level changes such as transposases, DNA-mediated transposons, acetyltransferases, and important membrane components. About 46% of the samples in group 2, which was strongly associated with the turquoise module, were from *V. cholerae* treated with different stresses, suggesting that *V. cholerae* adapted to challenges by regulating DNA-level changes and its membrane structure under different stress conditions. The functions enriched by hub genes within the blue module are mainly focused on chemotaxis, signal transduction, and metabolism-related pathways. Approximately 61% of the group 3 samples strongly associated with the blue module are from genetically manipulated *V. cholerae* which were subsequently re-stressed or entered the host, making it difficult to distinguish whether genetic manipulation or another factor is the main effector. The functions enriched by the hub genes in the brown module are

focused on cytoplasmic, signal transduction, and metabolic pathways of bacterial responses to different environments. About 63% of the samples in group 4, which is strongly associated with the brown module, are from *V. cholerae* cultured *in vitro*, suggesting that *V. cholerae* cultured under different conditions *in vitro* regulate metabolic pathways that are adapted to different environments through signal transduction.

DNA-mediated transposons and transposases are commonly associated with horizontal gene transfer in *V. cholerae*. Horizontal gene transfer is an important means for *V. cholerae* to rapidly acquire adaptive traits such as antibiotic resistance. Notably, *V. cholerae* possesses the ability of natural transformation on the surface of chitin in the natural water environment to take up exogenous DNA (24, 25). The transposon insertion activities of *V. cholerae* are involved in their acquisition of antibiotic resistance, phage resistance, and many other important adaptations (26). Chemotaxis and signal transduction systems are important in the trophic competition of bacteria, and signal transduction systems are thought to be important for regulation of gene expression in bacteria. In *V. cholerae*, chemotaxis not only recognizes nutrients and regulates the flagellar system, but is also involved in the regulation of virulence, biofilms, and other important colonization-related abilities (27). Interestingly, mutants of specific chemotaxis genes in the small intestines of infant mice exhibit a colonization advantage (28).

Histone-like nucleoid structuring protein (H-NS) is an important class of regulatory proteins in bacteria which affect the transcriptional functions of RNA polymerase by binding to DNA. H-NS is now known to have important functions within *Enterobacteriaceae* in the regulation of pathogenicity and multidrug resistance (29–31). In contrast, *toxR*-mediated antagonism of H-NS binding to DNA and thus driving intestinal colonization has also been identified in *V. cholerae* (32), so it is not surprising that in the hub gene expression pattern of group 3, the Δ*hns* strain was similar to the host expression profile. Previous studies (33) have shown that *oxyR1* and *oxyR2* regulate antioxidant enzymes in opposite directions, with the Δ*oxyR2* strain being more resistant to reactive oxygen species (ROS). However, in addition to its regulatory role in ROS resistance, *oxyR1* also acts on the two-component regulatory system *vc1084-vc1086*, CTX phage production (*vc1453-vc1454*, *vc1462-vc1463*). Considering that the Δ*oxyR1* strain was similar to human samples with or without $H_2O_2$, a correlation between the Δ*oxyR1* strain and human samples in terms of ROS resistance under the group 3 hub gene expression pattern can be ruled out, and it is possible that other downstream genes play a primary role. Tobramycin is a class of aminoglycoside antibiotics which cause double-stranded DNA breaks in *V. cholerae*. When *radD* or *recBCD* is absent, previous studies (34, 35) have also shown that tobramycin affects *V. cholerae* population-sensing-associated regulatory genes such as AI-2/LuxS QS and *hapR*, which mediate the SOS response in *V. cholerae*. The quorum-sensing signaling system of *V. cholerae* has also been shown in previous studies to play a key role in host intestinal colonization.

Many animal models are currently used to study *V. cholerae*, including infant mice, infant rabbits, *Drosophila*, adult mice, and amoebae. However, the different animal models only represent specific aspects of *V. cholerae* pathogenesis and do not fully simulate its real situation in human infections. In our study, we also found that different *in vivo* samples from different hosts are naturally assigned to different groups, where human samples are still unique. This implies that we still need to develop animal models which more closely resemble human infections to advance research related to *V. cholerae*.

Here, we explain the reasons for the method used within the manuscript. For RNA-seq expression data extraction, given the large number of samples included in this analysis, we chose Kallisto because it has the fastest processing speed, in line with the previous publications. For integration of microarray and RNA-seq, we used three currently published integration methods, including Rank-in, the Limma R package normalizedBetweenArrays, and TDM (36); Of these, Rank-in and the Limma R package normalizedBetweenArrays allow for data integration; however, example results were not obtained when the demo data was processed using the TDM package (data not shown), Rank-in and the Limma R package normalizedBetweenArrays function were chosen.

After integration using the Limma R package normalizedBetweenArrays function, the hierarchical clustering results showed that approximately 80% of the samples were grouped into the same group, exhibiting anomalous sample clustering (Fig. S4). We therefore conducted a WGCNA analysis of the integration data from the Limma R package normalizedBetweenArrays function pipeline using the clustering grouping information from the Rank-in pipeline. The results showed that more functional modules were identified than in the Rank-in pipeline, but the modules in red, yellow, brown, and turquoise still possessed some correlation. The correlations between modules and groups in both pipelines were consistent, i.e., group 2 with the turquoise module and group 3 with the blue module. Gene composition within both modules is shown in a Sankey diagram (Fig. S6G).

The data distribution of expression intensity processed by each method was checked and compared between array and RNA-seq technologies. The raw data without processing demonstrated two peaks in microarray but one completely deviating peak in RNA-seq (Fig. S5). In contrast to the large difference in the uncorrected method, the Limma R package normalizedBetweenArrays function compresses data from both platforms into one peak, and Rank-in tries to eliminate deviation peaks (Fig. S6B and C). For sample clustering, Rank-in presents a more uniform grouping, while the Limma R package normalizedBetweenArrays function has most of the samples clustered in group 1 (Fig. S4B). Combining the expression distribution, sample clustering, and independence between functional modules after integration of the two pipelines suggests that Rank-in may be more applicable to our data set than the Limma R package normalizedBetweenArrays function.

WGCNA analysis was performed using *in vivo* data with the *in vitro* data from the integrated data set. *In vitro* control samples from the three hosts were designated the *in vitro* group, and clinical human samples, infant rabbit samples, and *Drosophila* samples were divided into three other groups. The results showed that the functional modules enriched were not the same between hosts (Fig. S7C). These samples were simply divided into *in vivo* and *in vitro* data for WGCNA analysis, where the blue module exhibited a significant negative correlation with the *in vivo* group and the gray module exhibited a significant positive correlation (Fig. S7D). By comparing the KEGG-annotated *V. cholerae* infection-associated genes with the gray module genes, fewer genes were found to be shared, in agreement with other human data. Namely, the virulence-associated-genes did not seem to be significantly upregulated in the human gut.

## MATERIALS AND METHODS

**Data collection and processing.** All RNA-seq data were downloaded from the Sequence Read Archive (SRA) (9) and literature supplemental materials from PubMed. All microarray data were downloaded from the GEO database. Next, we converted RNA-seq raw data into compressed fastq files using the SRA Toolkit. Samples were mapped to a recently inferred *V. cholerae* transcriptome derived from the N16961 reference genome (37) using Kallisto version 0.45.1 (10). The two channels of the microarray data were treated as independent experiments, and we then used an in-house R script to perform the normalization. For RNA-seq, we selected a mapping rate of >50% as the cutoff to remove low-quality samples (5). This resulted in 382 samples being screened from 511 samples. For microarray, we collected 362 samples. Table S1 in the supplemental material contains details on the included experiments.

**Cross-platform data integration.** For Rank-in, the RNA-seq TPM generated by Kallisto was combined with the gene expression from the microarray into a gene expression matrix. Experimental grouping information for the corresponding samples was collated into a phenotype matrix. Both files were submitted to the Rank-in website to perform data integration.

For the Limma R package normalizedBetweenArrays function, the RNA-seq count generated by Kallisto was combined with the gene expression from the microarray into a gene expression matrix. The entire gene matrix was homogenized in the R environment using the normalizedBetweenArrays directive in the Limma package.

**WGCNA.** A weighted gene co-expression network analysis was performed using WGCNA (https://github.com/ShawnWx2019/WGCNA-shinyApp) by entering the gene expression matrix. This process consists of sequential calculation of a Pearson correlation matrix, adjacency matrix with power $\beta = 3$, and ultimately a topological overlap matrix (TOM) (38) from normalized gene expression counts across conditions. We further filtered this TOM to exclude samples with weighted coexpression of <0.1 for all analyses included in the results.

Predicted pathway annotations and gene functional knowledge were derived from the NCBI Biosystems database as well as the DAVID, GO, and KEGG databases (39–41).

**Data availability.** SRA and GEO accession numbers and information on all included samples can be found in Table S1 in the supplemental material. The integral gene expression matrix is provided in Tables S3 and S4. The PC list table is provided in Table S2. Sample clustering group information list table is provided in Table S5. All data analysis and figure generation was performed using the Linux and R programming language. All data relevant to the study are included in the article or have been uploaded as supplementary information.

## SUPPLEMENTAL MATERIAL

Supplemental material is available online only.
**SUPPLEMENTAL FILE 1**, PDF file, 3.3 MB.
**SUPPLEMENTAL FILE 2**, PDF file, 0.2 MB.
**SUPPLEMENTAL FILE 3**, PDF file, 0.1 MB.
**SUPPLEMENTAL FILE 4**, XLSX file, 28.9 MB.
**SUPPLEMENTAL FILE 5**, XLSX file, 12.3 MB.
**SUPPLEMENTAL FILE 6**, XLSX file, 0.01 MB.

## ACKNOWLEDGMENTS

This work was supported by the National Natural Science Foundation of China (31770132, 81572050 and 81873969).

Z-X.Q., G-Z.C., W-H.C., and Z.L. designed and finalized the study. Z-X.Q., G-Z.C., Y-J.W., and Q-Q.Y. conducted bioinformatics analysis. Z-X.Q., G-Z.C., X-M.Y., C-Q.S., and M.L. helped with data collection. Z-X.Q. drew the charts and wrote the paper. J.Z., W-H.C., and Z.L. supervised the project. All the authors approved the final version submitted.

We have no competing interests to declare.

This study was not commissioned and was externally peer reviewed.

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
