## [Reviewer comments · Microbiology Spectrum]

Microbiology Spectrum

Cross-platform transcriptomic data integration, profiling and mining in *Vibrio cholerae*

Zi-Xin Qin, Guozhong Chen, Qian-Qian Yang, Ying-Jian Wu, Chuqing Sun, Xiaoman Yang, Mei Luo, Chunrong Yi, Zhu Jun, Wei-Hua Chen, and Zhi Liu

Corresponding Author(s): Wei-Hua Chen, Huazhong University of Science and Technology

Review Timeline:

Submission Date:	December 31, 2022
Editorial Decision:	February 12, 2023
Revision Received:	April 14, 2023
Accepted:	April 24, 2023

Editor: Beile Gao

Reviewer(s): The reviewers have opted to remain anonymous.

Transaction Report:

DOI: <https://doi.org/10.1128/spectrum.05369-22>

February 12, 2023

Prof. Wei-Hua Chen
Huazhong University of Science and Technology
Department of Bioinformatics and Systems Biology, College of Life Science and Technology
Luoyu Road 1037
Wuhan 430074
China

Re: Spectrum05369-22 (Cross-platform transcriptomic data integration, profiling and mining in *Vibrio cholerae*)

Dear Prof. Wei-Hua Chen:

Experts in the field have carefully reviewed your manuscript entitled " Cross-platform transcriptomic data integration, profiling and mining in *Vibrio cholerae*". Although they found merit in your study, they have raised a number of serious concerns that preclude its acceptance in the present form. This paper has been recommended for Modifications. I invite you to respond to the reviewers' comments and revise your paper accordingly. In addition, the writing should be improved to aid the reader in understanding the rationale for the analyses performed. The revised version will be re-evaluated by the original reviewers. Please be aware that this invitation to revise your manuscript does not guarantee eventual acceptance of your manuscript.

Link Not Available

Sincerely,

Beile Gao

Journals Department
Reviewer comments:

Reviewer #1 (Comments for the Author):

In this manuscript, the authors use bioinformatics methods to integrate all of the existing *V. cholerae* microarray and RNA-seq datasets in order to assess more global expression patterns by these bacteria. Through this analysis, they made some key

findings on genes whose expression are most active or are generally silent. A potentially helpful finding for the research community is their investigation of genes that are often used as expression controls for transcription studies. Their analysis suggests that many of the currently used genes may not have stable expression under different conditions, while suggesting alternative genes that may be more suitable for such studies. However, their clustering analysis is confusing, and it is not clear how relevant such analysis is without some type of validation studies. Overall, this work may provide the *V. cholerae* research community with some new insights, but more clarification on its utility would be helpful for most readers.

Specific points:

1. The relevance and utility of the clustering analysis is not well-articulated in the manuscript. It is difficult for the reader to understand why this is important and what it can be used to demonstrate. In addition, without any validation studies or follow-up it is unclear if these delineations are biologically relevant.
2. There are a large number of abbreviations in this study and most are not spelled out the first time they are used (lines 62, 106, 127, 260, 521, 528, Figure 1).
3. The rationale for the programs used for analysis could be better articulated.
4. Line 248 describes results from "mammary rats?"-this study was on infant mice.
5. The paragraph starting on line 236 is material that should not be included in the results section-it would be better suited to the discussion.

Reviewer #3 (Comments for the Author):

In this study Zi-Xin Qin and colleagues applied two approaches to integrate two types of *Vibrio cholerae* gene expression datasets: RNA-seq and microarray. This analysis using one of the approaches allowed the authors to find top genes consistently upregulated and downregulated across studies and platforms. Using hierarchical clustering the authors were able to cluster all the samples into 4 groups. Using the weighted gene correlation networks analysis applied to *V. cholerae* genes across studies the authors produced 6 gene modules. Next, they cross-related 4 groups with these 6 modules and established that the groups have correlations with certain modules. The protein-protein interaction analysis using STRING database showed that the genes from the three of the four groups have multiple interaction with each other, but the genes from one of the groups (group 1) did not show any significant interactions between each other. This prompted the authors to speculate that the cross-platform integration and grouping of samples allowed them to find a group of genes that can potentially interact directly or indirectly but that were never shown previously to be interacting.

First of all, the manuscript requires editing by a person with full professional proficiency in English.

The authors performed the analysis across 873 samples, which is impressive. But they did not attempt to propose how the results of this work can be beneficial for other studies. Could the authors elaborate on how they results can be used in other studies? What the identified 4 clusters and 6 modules can be used for?

Although the introduction states that two approaches were used to integrate RNA-seq and microarray data, the results of only one of them were actually described. The other approach, the so-called rank-in algorithm, is mentioned only in the discussion section. I would encourage the authors to show clearer how the rank-in approach compares to the approach using the Limma R package `normalizedBetweenArrays` function.

In the last paragraph of the discussion the authors mentioned that the rank-in algorithm outperforms the approach using the `normalizedBetweenArrays` algorithm. No rigorous analysis has been done though to make such a conclusion. Could the authors make a more detailed comparison and provide metrics that are used to compare the two approaches? In addition, can the authors provide an independent way to assess the correctness of the integration done across the platforms (considering that the two approaches gave somewhat different results)?

In the "PPI interaction networks between hubgene suggest novel interactions" section the authors just showed the protein-protein interaction network figures taken from STRING database and did not provide any kind of analysis. What insights did the networks give? What conclusion can be made based on their analysis?

I would encourage the authors to add to their GitHub repository the actual working code that was used to do the analysis. This will help readers to replicate and expand on the analysis.

Figures S2 and S4 are of poor quality. Please provide figures of better quality.
Figures 2 A and C are also not of best quality.

Specific comments:

Throughout the manuscript "hubgene" should be written separately and following the rules of plural/singular.

Line 54 - 'such as tcp' - explain in a couple of words what the gene does

Line 56 - 2011investigated - divide by space

Line 56 - CT, TCP - explain in the text what these abbreviations stand for

Line 62 - define the 'WGCNA' abbreviation and briefly explain how does the method work.

Line 68 - "is a big challenge"

Line 73 - Limma-normalizedBetweenArrays. Change this to "the Limma R package normalizedBetweenArrays function" throughout the manuscript. When referring to this as an approach it is better to give it a different name, as the package initially was developed to work with microarray data, but has been adapted in the paper cited by the authors (Castillo, D., et al., 2017) to normalize expression values across RNA-seq and microarray.

Line 97 - "is shown"

Line 101 - "... therefore they could not be used ..."

Line 106 - Rtsne is just a function in R. Explain what the 'Rtsne analysis' actually is.

Line 107 - same as in line 101.

Line 147 - vc2699 is indeed homologous to the E.coli transporter DcuA, but the referenced paper investigated not the transporter but another protein, C4-Dicarboxylate/citrate sensor kinase DcuS, which is a component of two-component system. Neither vc2699 nor its E.coli homolog is a component of two-component system. Correction is needed.

Line 181 - "We found that vc0992 and vc1670 performed well under both criteria (Fig. 3F)". This seem to contradict to the previous sentence, which states that vc0092 along with other genes was excluded. Please clarify.

Line 187 - The authors probably meant Fig.S2. The quality of the figure is low though and it is unreadable. A better quality figure is needed. Here also explain what hCluster stands for. Call it what it is - hierarchical clustering.

Line 204-206 - Restate these two sentences. They meaning is not clear.

Line 207-209 - Restate or divide into two sentences.

Line 207-222 - This is not entirely surprising as removing the group driving categories would expectedly lead to the weakening of the considered correlations.

Line 227 - Explain in the text what GS stand for.

Line 227 - Why the hub genes of the first module were not obtained?

Line 228 - Explain what MM stands for.

Line 238 - "to acquire adaptive evolution" is not a correct phrase.

Line 239 - What is "a natural receptor state"? The entire sentence (lines 239-242) is poorly written. Please restate.

Line 248 - What are "mammary rats"? Furthermore, a mouse model was used in the referenced paper.

Lines 251-253 - The sentence does not read naturally. Please restate.

Lines 206, 257 - The authors probably meant "simulation" here not "stimulation". But even so, I do not think simulation is a correct description of in vitro experiments or in vivo but in model organisms. They are just that, in vitro experiments and in vivo in model organisms, through which researchers make hypotheses about processes in the human organism, that can be later confirmed or refused. So, I would suggest not calling in vitro or in vivo experiments in model organisms simulations. The word "simulate" is also used in line 352. Correct all instances of calling in vitro or in vivo experiments in model organisms simulation.

Line 260 - First define the abbreviation "PCA".

Line 266-267 - Drop "principal component analysis" as you used its abbreviation right before these words.

Line 274 - "PPI interaction networks between hubgene suggest novel interactions". Why the PPI network was not constructed for the gray module (Group 1)?

Line 302 - The authors apparently meant Figure S4 here. Please correct this and also check figures numbering throughout the manuscript.

Line 357-366 - This paragraph is very poorly written. Please carefully rewrite it and show clearer what message it conveys.

Figures S2 and S3 have only titles but no details. Please provide details after the titles.

Figure S4E - Indicate in the figure which integration approach is on the left and which is on the right.

Tables S1 and S2 do not have titles. Please provide the titles.

I have unpacked the rar archives with the Table S3 parts but could not open them. "The files is corrupted" message popped up when I tried opening the files. Please check the files.

Staff Comments:

Preparing Revision Guidelines

Please return the manuscript within 60 days; if you cannot complete the modification within this time period, please contact me. If you do not wish to modify the manuscript and prefer to submit it to another journal, please notify me of your decision immediately so that the manuscript may be formally withdrawn from consideration by Microbiology Spectrum.

In this study Zi-Xin Qin and colleagues applied two approaches to integrate two types of *Vibrio cholerae* gene expression datasets: RNA-seq and microarray. This analysis using one of the approaches allowed the authors to find top genes consistently upregulated and downregulated across studies and platforms. Using hierarchical clustering the authors were able to cluster all the samples into 4 groups. Using the weighted gene correlation networks analysis applied to *V. cholerae* genes across studies the authors produced 6 gene modules. Next, they cross-related 4 groups with these 6 modules and established that the groups have correlations with certain modules. The protein-protein interaction analysis using STRING database showed that the genes from the three of the four groups have multiple interaction with each other, but the genes from one of the groups (group 1) did not show any significant interactions between each other. This prompted the authors to speculate that the cross-platform integration and grouping of samples allowed them to find a group of genes that can potentially interact directly or indirectly but that were never shown previously to be interacting.

First of all, the manuscript requires editing by a person with full professional proficiency in English.

The authors performed the analysis across 873 samples, which is impressive. But they did not attempt to propose how the results of this work can be beneficial for other studies. Could the authors elaborate on how they results can be used in other studies? What the identified 4 clusters and 6 modules can be used for?

Although the introduction states that two approaches were used to integrate RNA-seq and microarray data, the results of only one of them were actually described. The other approach, the so-called rank-in algorithm, is mentioned only in the discussion section. I would encourage the authors to show clearer how the rank-in approach compares to the approach using the Limma R package `normalizedBetweenArrays` function.

In the last paragraph of the discussion the authors mentioned that the rank-in algorithm outperforms the approach using the `normalizedBetweenArrays` algorithm. No rigorous analysis has been done though to make such a conclusion. Could the authors make a more detailed comparison and provide metrics that are used to compare the two approaches? In addition, can the authors provide an independent way to assess the correctness of the integration done across the platforms (considering that the two approaches gave somewhat different results)?

In the “PPI interaction networks between hubgene suggest novel interactions” section the authors just showed the protein-protein interaction network figures taken from STRING database and did not provide any kind of analysis. What insights did the networks give? What conclusion can be made based on their analysis?

I would encourage the authors to add to their GitHub repository the actual working code that was used to do the analysis. This will help readers to replicate and expand on the analysis.

Figures S2 and S4 are of poor quality. Please provide figures of better quality.

Figures 2 A and C are also not of best quality.

Specific comments:

Throughout the manuscript “hubgene” should be written separately and following the rules of plural/singular.

Line 54 – ‘such as tcp’ – explain in a couple of words what the gene does

Line 56 – 2011investigated – divide by space

Line 56 – CT, TCP – explain in the text what these abbreviations stand for

Line 62 – define the ‘WGCNA’ abbreviation and briefly explain how does the method work.

Line 68 – “is a big challenge”

Line 73 – Limma-normalizedBetweenArrays. Change this to “the Limma R package normalizedBetweenArrays function” throughout the manuscript. When referring to this as an approach it is better to give it a different name, as the package initially was developed to work with microarray data, but has been adapted in the paper cited by the authors (Castillo, D., et al., 2017) to normalize expression values across RNA-seq and microarray.

Line 97 – “is shown”

Line 101 – “... therefore they could not be used ...”

Line 106 – Rtsne is just a function in R. Explain what the ‘Rtsne analysis’ actually is.

Line 107 – same as in line 101.

Line 147 – vc2699 is indeed homologous to the E.coli transporter DcuA, but the referenced paper investigated not the transporter but another protein, C4-Dicarboxylate/citrate sensor kinase DcuS, which is a component of two-component system. Neither vc2699 nor its E.coli homolog is a component of two-component system. Correction is needed.

Line 181 – “We found that vc0992 and vc1670 performed well under both criteria (Fig. 3F)”. This seem to contradict to the previous sentence, which states that vc0092 along with other genes was excluded. Please clarify.

Line 187 – The authors probably meant Fig.S2. The quality of the figure is low though and it is unreadable. A better quality figure is needed. Here also explain what hCluster stands for. Call it what it is - hierarchical clustering.

Line 204-206 – Restate these two sentences. They meaning is not clear.

Line 207-209 – Restate or divide into two sentences.

Line 207-222 – This is not entirely surprising as removing the group driving categories would expectedly lead to the weakening of the considered correlations.

Line 227 – Explain in the text what GS stand for.

Line 227 – Why the hub genes of the first module were not obtained?

Line 228 – Explain what MM stands for.

Line 238 - “to acquire adaptive evolution” is not a correct phrase.

Line 239 – What is “a natural receptor state”? The entire sentence (lines 239-242) is poorly written. Please restate.

Line 248 – What are “mammary rats”? Furthermore, a mouse model was used in the referenced paper.

Lines 251-253 – The sentence does not read naturally. Please restate.

Lines 206, 257 – The authors probably meant “simulation” here not “stimulation”. But even so, I do not think simulation is a correct description of *in vitro* experiments or *in vivo* but in model organisms. They are just that, *in vitro* experiments and *in vivo* in model organisms, through which researchers make hypotheses about processes in the human organism, that can be later confirmed or refused. So, I would suggest not calling *in vitro* or *in vivo* experiments in model organisms

simulations. The word “simulate” is also used in line 352. Correct all instances of calling *in vitro* or *in vivo* experiments in model organisms simulation.

Line 260 – First define the abbreviation “PCA”.

Line 266-267 - Drop “principal component analysis” as you used its abbreviation right before these words.

Line 274 - “PPI interaction networks between hubgene suggest novel interactions”. Why the PPI network was not constructed for the gray module (Group 1)?

Line 302 – The authors apparently meant Figure S4 here. Please correct this and also check figures numbering throughout the manuscript.

Line 357-366 – This paragraph is very poorly written. Please carefully rewrite it and show clearer what message it conveys.

Figures S2 and S3 have only titles but no details. Please provide details after the titles.

Figure S4E – Indicate in the figure which integration approach is on the left and which is on the right.

Tables S1 and S2 do not have titles. Please provide the titles.

I have unpacked the rar archives with the Table S3 parts but could not open them. “The files is corrupted” message popped up when I tried opening the files. Please check the files.

Responses to reviewer #1:

Comments/questions #1:

The relevance and utility of the clustering analysis is not well-articulated in the manuscript. It is difficult for the reader to understand why this is important and what it can be used to demonstrate. In addition, without any validation studies or follow-up it is unclear if these delineations are biologically relevant.

Response:

Thank you for your comments.

Hierarchical clustering analysis is a method often used in omics studies to assess the consistency of samples within groups and the variability of component samples. In transcriptomes, high inter-sample similarity represents similarity in transcriptional profiles between samples, i.e., similar expression changes/levels of specific genes [1]. In cross-platform data integration, clustering analysis is also used to assess the reliability of integrated data [2, 3]. *In vitro* experiments of pathogenic bacteria are expected to simulate infection in humans as much as possible. Unfortunately, transcriptome studies of *Vibrio cholerae* in clinical patients mainly generated from microarray, while RNA-seq is mainly used in laboratory samples. And the principles of these two techniques are different, resulting in the inability to compare laboratory data with data from precious clinical samples. In this manuscript, we make it possible to compare these two data through data integration, and then use conventional clustering methods to evaluate those laboratory conditions that are similar to those of clinical samples. Therefore, based on your suggestion, we reveal its meaning and applicability in detail in our article, as detailed in our actions part.

We agree with you that additional validation is important. We first performed clustering grouping by transcriptional profiles of all genes and found that all experimental samples could be divided into 4 groups and human samples were clustered into the first group. Then, we wanted to find laboratory samples that shared expression characteristics with human samples to provide research directions for

laboratory studies. However, by the full genes analysis, the human samples were far away from all the experimental samples, even if they were in the same group. Since hub gene was considered by the data analysis as a key factor in the classification of the different groups, we used hub gene as an indicator to re-cluster our data. The result shows that the gene expression profile of Δhns strains and $\Delta oxyR1$ strains were closer to humans in group3 (Figure 6B), and in group4, the tobramycin-treated samples is close to human samples (Figure 6C), suggesting that the expression of these genes may be associated with *Vibrio cholerae* infection in humans. To validate this conclusion, we used newly sourced microarray samples of *Vibrio cholerae* infecting humans and showed that 50% of the genes (5/10) in the Δhns strains were also found in the new human samples (Revised Figure 6D) and ~35% of the genes (20/56) in the tobramycin samples were also found in humans (Revised Figure 6E). This demonstrates the reliability of our analytical approach, although eventually a large number of biological experiments will be needed to verify the role of these genes in the colonization of *Vibrio cholerae*. Also, we performed validation using *in vivo* data, dividing human samples, infant rabbit samples and *drosophila* samples into three groups, with the corresponding *in vitro* control samples serving as the *in vitro* group. WGCNA revealed no identical highly correlated modules, corresponding to these three *in vivo* samples being assigned to different groups (Revised Figure S7).

1. Petegrosso, R., Z. Li, and R. Kuang, *Machine learning and statistical methods for clustering single-cell RNA-sequencing data*. Brief Bioinform, 2020. **21**(4): p. 1209-1223.
2. Tang, K., et al., *Rank-in: enabling integrative analysis across microarray and RNA-seq for cancer*. Nucleic Acids Res, 2021. **49**(17): p. e99.
3. Castillo, D., et al., *Integration of RNA-Seq data with heterogeneous microarray data for breast cancer profiling*. BMC Bioinformatics, 2017. **18**(1): p. 506.

Action:

- We have added detailed information to illustrate the importance of this work in line 77-82.
- We added the results of the validation work in Figure 6D, 6E and S7.

The old figure:

Figure 6

The revised figure:

Figure 6

Figure S7

Comments/questions #2:

There are a large number of abbreviations in this study and most are not spelled out the first time they are used (lines 62, 106, 127, 260, 521, 528, Figure 1).

Response:

We are thankful for this valuable comment of the reviewer. All these abbreviations are spelled out the first time they are used in new version.

Action:

- We have added these abbreviations in new manuscript.

Comments/questions #3:

The rationale for the programs used for analysis could be better articulated.

Response:

We are thankful to the reviewer for this comment.

For RNA-seq expression data extraction, given the large number of samples included in this analysis, kallisto was chosen as the fastest processing speed, in line with the previous publications. For integration of microarray and RNA-seq, there are currently three published articles that achieve integration, including Rank-in, the Limma R package normalizedBetweenArrays function and TDM, and we used all three methods simultaneously to attempt integration. Of these, Rank-in and the Limma R package normalizedBetweenArrays function allow for data integration,

however, the example results were not obtained when the demo data was processed using the TDM package (not shown in this paper), so Rank-in and the Limma R package normalizedBetweenArrays function were chosen.

Action:

- We have added detailed information to the rationale for the programs used for analysis in line 389-398.

Comments/questions #4:

Line 248 describes results from "mammary rats?"-this study was on infant mice.

Response:

We are thankful to the reviewer for this comment. This error has been corrected in the new version of the manuscript.

Actions:

- We have corrected this error in new manuscript.

Comments/questions #5:

The paragraph starting on line 236 is material that should not be included in the results section-it would be better suited to the discussion.

Response:

We are thankful for this valuable comment of the reviewer. In the new version of this manuscript, this part has been moved to the discussion section.

Actions:

- We have moved this part to line 346.

Responses to reviewer #2:

Comments/questions #1:

First of all, the manuscript requires editing by a person with full professional proficiency in English.

Response:

We are thankful to the reviewer for this comment. This manuscript has been reviewed by a native speaker.

Comments/questions #2:

The authors performed the analysis across 873 samples, which is impressive. But they did not attempt to propose how the results of this work can be beneficial for other studies. Could the authors elaborate on how they results can be used in other studies? What the identified 4 clusters and 6 modules can be used for?

Response:

We are thankful for this valuable comment of the reviewer.

Firstly, we compared the expression of the internal reference genes at an overall level and obtained candidate internal reference genes with potential applications. The internal reference genes identified in this paper are both more stable and moderately expressed than currently available internal reference genes, and can be applied in the study of *Vibrio cholerae* gene expression. Also, our cross-platform analysis has identified a large class of previously unknown functional genes associated with transposases that can be used in bacterial transposition studies. Importantly, Through analysis we found that Δhns , $\Delta oxyR1$, tobramycin partial gene expression patterning is consistent with human gene expression, and these genes may be important candidates for the study of *Vibrio cholerae* infection process. Meanwhile, the integration of data also allowed for downstream univariate analysis, such as comparison of data between different hosts. Data analysis revealed that gene expressions in infant rabbits or mice are still very different from human data, and other new models may need to be developed.

The four clusters are formed by the similarity of gene expression across all integrated samples, with objects in the same cluster having similar characteristics and those in different clusters having different characteristics. In the context of gene expression studies, either gene-based or sample-based clustering can be applied. Gene-based clustering is the process of grouping genes according to their expression values in different treatment groups. Such clustering analysis helps to identify groups of genes that are co-expressed, and co-expressed genes are often functionally related [4]. In our analysis, clustering analysis allows us to explore what exactly is the

biological significance of the effect of different treatment conditions on *Vibrio cholerae*. Modules are based on co-expression relationships, and genes in the same module mean that these genes are co-expressed, i.e. they may be functionally related. For example, genes within the turquoise module may also be involved in transposition activities.

4. Rawat, A., G.J. Seifert, and Y. Deng, *Novel implementation of conditional co-regulation by graph theory to derive co-expressed genes from microarray data*. BMC Bioinformatics, 2008. **9 Suppl 9**(Suppl 9): p. S7.

Comments/questions #3:

Although the introduction states that two approaches were used to integrate RNA-seq and microarray data, the results of only one of them were actually described. The other approach, the so-called rank-in algorithm, is mentioned only in the discussion section. I would encourage the authors to show clearer how the rank-in approach compares to the approach using the Limma R package normalizedBetweenArrays function.

Response:

Thanks to your suggestion, we performed an integrated analysis of the *Vibrio cholerae* data using both methods, and the WGCNA results were broadly similar. As the Limma R package normalizedBetweenArrays function clustering does not fit reasonably (see Figure S4), we mainly show the rank-in approach in the main text and therefore place the Limma R package normalizedBetweenArrays function data in the supplementary material for the reader interested in limma. For ease of understanding, the data for our two comparisons are placed in the Discussion section (line 392-414).

Actions:

- We have added additional information to the discussion section, corresponding to the figure in Figure S5, S6 and S7.

The old figure:

Figure S4

The revised figure:
Figure S4

Figure S5

Figure S6

Comments/questions #4:

In the last paragraph of the discussion the authors mentioned that the rank-in algorithm outperforms the approach using the normalizedBetweenArrays algorithm. No rigorous analysis has been done though to make such a conclusion. Could the authors make a more detailed comparison and provide metrics that are used to compare the two approaches? In addition, can the authors provide an independent way to assess the correctness of the integration done across the platforms (considering that the two approaches gave somewhat

different results)?

Response:

Thank you for your suggestion. Based on your suggestion, we have done a comparative analysis of the transcript level distribution and sample distribution of the two integrated data. Based on our data (Figure S4, S5 and S6), the samples of the limma integrated data are not evenly distributed, so we present the results using Rank-in in the main text, and also put the limma results in the supplementary material for interested readers, but does not imply a superiority or inferiority between the two.

Most of the conclusions are in agreement, proving that the analysis has some reliability. We also wanted to find a third method for further independent validation, and there are currently others applying the TDM method for cross-platform integration, which we attempted, but unfortunately we used the demo data from this R package for our calculations and obtained results that did not agree with the demo results.

Actions:

- We have added additional information to the discussion section, corresponding to the figure in Figure S4, S5 and S6.

TDM demo:

If we TDM transform the data, the results appear to be much improved:

```
tcga_tdm = tdm_transform(ref_data = data.table(cbind(gene=rownames(meta), meta)),
target_data = data.table(cbind(gene=rownames(tcga), tcga)))
summary(as.vector(data.matrix(tcga_tdm[,2:ncol(tcga_tdm),with=F])))

##      Min. 1st Qu.  Median    Mean 3rd Qu.    Max.
## 4.776  6.244  7.543  7.684  8.869 14.880
```

Finally, here is a plot comparing the distributions of the reference data, the scaled data, the log transformed data, and the TDM transformed data:

Our results:

```
> summary(as.vector(data.matrix(tcga_tdm[,2:ncol(tcga_tdm),with=F])))
      Min. 1st Qu.  Median    Mean 3rd Qu.    Max.
      1      477     952    1015    1525    2364
```

Comments/questions #5:

In the "PPI interaction networks between hubgene suggest novel interactions" section the authors just showed the protein-protein interaction network figures

taken from STRING database and did not provide any kind of analysis. What insights did the networks give? What conclusion can be made based on their analysis?

Response:

Thank you for your suggestion, our presentation was not clear. Following your suggestion, we mapped the PPI interworking network by hub genes mining for three modules (see Figure 7). We then entered the same gene number into the STRING database and also obtained a PPI relationship graph for the STRING source (see Figure S3). By comparing the PPI interworking graphs of the two sources (see Figure 7), we found that the PPI relationships between the blue module and the hub gene within the brown module shared a specific percentage (~10-20%) with the STRING source. Interestingly, we found that only 0.26% of the PPIs in the turquoise module could be consistent with the PPIs of the STRING source (see Figure 7B). In particular, the genes with the most PPI in the turquoise Module are mainly concentrated in transposases, implying that we have found transposase-related gene functional modules.

Actions:

- We have revised the PPI network based on the WGCNA co-expression relationship and compared it with the PPI network from STRING sources. Detailed information is added in lines 277-297, corresponding to Figure 7 and Figure S3.

The old figure:

Figure 7

The revised figure:

Figure 7

Figure S3

Comments/questions #6:

I would encourage the authors to add to their GitHub repository the actual working code that was used to do the analysis. This will help readers to replicate and expand on the analysis.

Response:

Thanks for your comment. The working code has been added to the GitHub repository (address: [ZixinQin/Dataset-for-Cross-platform-transcriptomic-data](https://github.com/ZixinQin/Dataset-for-Cross-platform-transcriptomic-data) at [Update \(github.com\)](https://github.com/ZixinQin/Dataset-for-Cross-platform-transcriptomic-data)).

Comments/questions #7:

Figures S2 and S4 are of poor quality. Please provide figures of better quality.

Figures 2 A and C are also not of best quality.

Response:

Thanks for your comment. This problem occurs when the image containing the sample name is itself too large to be combined with other images and then exported as a single file. Revised ew figure has been added to the manuscript. We have also provided a PDF file for readers to review.

Revised figure detail (PDF file):

Comments/questions #8:

Throughout the manuscript "hubgene" should be written separately and following the rules of plural/singular.

Response:

Thanks for your comment. All "hubgene" have been corrected to "hub gene".

Comments/questions #9:

Line 54 - 'such as tcp' - explain in a couple of words what the gene does

Response:

Thanks for your comment. "tcp" means the toxin co-regulated pilus gene family, which were identified as a critical colonization factor in both animal models and humans for *Vibrio cholerae*. The details have been added to the manuscript.

Comments/questions #10:

Line 56 - 2011investigated - divide by space

Response:

Thanks for your comment. We have fixed this error.

Comments/questions #11:

Line 56 - CT, TCP - explain in the text what these abbreviations stand for

Response:

Thanks for your comment. Cholera toxin (CT) and toxin co-regulated pilus (TCP) are the critical colonization factors of *Vibrio cholerae* in gut. The details have been added to the manuscript.

Comments/questions #12:

Line 62 - define the 'WGCNA' abbreviation and briefly explain how does the method work.

Response:

Thanks for your comment. Weighted correlation network analysis (WGCNA) is a popular method for detecting and investigating highly correlated gene clusters based upon high-throughput sequencing expression datasets. The details have been added to the manuscript.

Comments/questions #13:

Line 68 - "is a big challenge"

Response:

Thanks for your comment. We have fixed this error.

Comments/questions #14:

Line 73 - Limma-normalizedBetweenArrays. Change this to "the Limma R package normalizedBetweenArrays function" throughout the manuscript. When referring to this as an approach it is better to give it a different name, as the package initially was developed to work with microarray data, but has been adapted in the paper cited by the authors (Castillo, D., et al., 2017) to normalize expression values across RNA-seq and microarray.

Response:

Thanks for your comment. All “Limma-normalizedBetweenArrays” have been corrected to “the Limma R package normalizedBetweenArrays function”.

Comments/questions #15:

Line 97 - "is shown"

Response:

Thanks for your comment. We have fixed this error.

Comments/questions #16:

Line 101 - "... therefore they could not be used ..."

Response:

Thanks for your comment. We have fixed this error.

Comments/questions #17:

Line 106 - Rtsne is just a function in R. Explain what the 'Rtsne analysis' actually is.

Response:

Thanks for your comment. We have used tsne only for data downscaling observations, which is not properly called analysis and has been revised in the manuscript.

Comments/questions #18:

Line 107 - same as in line 101.

Response:

Thanks for your comment. We have fixed this error.

Comments/questions #19:

Line 147 - vc2699 is indeed homologous to the E.coli transporter DcuA, but the referenced paper investigated not the transporter but another protein, C4-Dicarboxylate/citrate sensor kinase DcuS, which is a component of two-component system. Neither vc2699 nor its E.coli homolog is a component of two-component system. Correction is needed.

Response:

Thanks for your comment. We have correctly cited the literature studying DcuA in line 160.

Comments/questions #20:

Line 181 - "We found that vc0992 and vc1670 performed well under both criteria (Fig. 3F)". This seem to contradict to the previous sentence, which states that vc0092 along with other genes was excluded. Please clarify.

Response:

Thanks for your comment. In the previous sentence, it was *vc0092* and other genes that were excluded. In Figure 3F it was *vc0992*, not *vc0092*.

Comments/questions #21:

Line 187 - The authors probably meant Fig.S2. The quality of the figure is low though and it is unreadable. A better quality figure is needed. Here also explain what hCluster stands for. Call it what it is - hierarchical clustering.

Response:

Thanks for your comment. We have retried the hierarchical clustering and found that it is still poorly readable, so we have provided the sample grouping information as an additional table for the reader to review.

Actions:

- We have added additional Table S5.

Comments/questions #22:

Line 204-206 - Restate these two sentences. Their meaning is not clear.

Response:

Thanks for your comment. We have rephrased these two sentences.

Comments/questions #23:

Line 207-209 - Restate or divide into two sentences.

Response:

Thanks for your comment. We have rephrased these two sentences.

Comments/questions #24:

Line 207-222 - This is not entirely surprising as removing the group driving categories would expectedly lead to the weakening of the considered correlations.

Response:

Thanks for your comment. We agree with you. The extraction of some data leads to a decrease in correlation, but the degree of decrease varies. And we believe that the more declining sample types contribute more to the correlation between modules and groups. For example (See Figure 4D), where the correlation between Group2 and the turquoise module decreases significantly when the stress conditional samples are

removed, while the correlation between group3 and group4 and the corresponding modules does not change significantly. We believe that the significant decrease may suggest a dominant role of stress samples in this correlation.

Comments/questions #25:

Line 227 - Explain in the text what GS stand for.

Response:

Thanks for your comment. Gene significance (GS) and Module Membership (MM) are criteria used in the WGCNA pipeline to determine the hub gene. The details have been added to the manuscript.

Comments/questions #26:

Line 227 - Why the hub genes of the first module were not obtained?

Response:

Thanks for your comment. In WGCNA, the genes within the grey modules are genes that are not classified into other modules and therefore do not have a co-expression relationship in themselves. The details is in line 220-221.

Comments/questions #27:

Line 228 - Explain what MM stands for.

Response:

Thanks for your comment. Gene significance (GS) and Module Membership (MM) are criteria used in the WGCNA pipeline to determine the hub gene. The details have been added to the manuscript.

Comments/questions #28:

Line 238 - "to acquire adaptive evolution" is not a correct phrase.

Response:

Thanks for your comment. We have rephrased it to “Horizontal gene transfer is an important means for *V. cholerae* to rapidly acquire adaptive traits such as antibiotic resistance”.

Comments/questions #29:

Line 239 - What is "a natural receptor state"? The entire sentence (lines 239-242) is poorly written. Please restate.

Response:

Thanks for your comment. Natural competent state should be more suitable. This is the state in which *Vibrio cholerae* can naturally transform to take up exogenous DNA on the surface of chitin. We have rephrased these sentences.

Comments/questions #30:

Line 248 - What are "mammary rats"? Furthermore, a mouse model was used in the referenced paper.

Response:

Thanks for your comment. This should be "Infant mice". We have fixed this error.

Comments/questions #31:

Lines 251-253 - The sentence does not read naturally. Please restate.

Response:

Thanks for your comment. We have rephrased these sentences.

Comments/questions #32:

Lines 206, 257 - The authors probably meant "simulation" here not "stimulation". But even so, I do not think simulation is a correct description of in vitro experiments or in vivo but in model organisms. They are just that, in vitro experiments and in vivo in model organisms, through which researchers make hypotheses about processes in the human organism, that can be later confirmed or refused. So, I would suggest not calling in vitro or in vivo experiments in model organisms simulations. The word "simulate" is also used in line 352. Correct all instances of calling in vitro or in vivo experiments in model organisms simulation.

Response:

Thanks for your comment. we have carefully considered the comment and believe that the use of stimulation once may not be appropriate. In this manuscript we focus on the transcriptome of pathogenic bacteria *in vivo* and *in vitro*, where the *in vitro* environment is subdivided into those with and without stress treatment. This part of the description will therefore be described in detail. All incorrect descriptions have

also been fixed.

Comments/questions #33:

Line 260 - First define the abbreviation "PCA".

Response:

Thanks for your comment. We have fixed this error.

Comments/questions #34:

Line 266-267 - Drop "principal component analysis" as you used its abbreviation right before these words.

Response:

Thanks for your comment. We have fixed this error.

Comments/questions #35:

Line 274 - "PPI interaction networks between hubgene suggest novel interactions". Why the PPI network was not constructed for the gray module (Group 1)?

Response:

Thanks for your comment. In WGCNA, the genes within the grey modules are not classified into other modules and therefore do not have a co-expression relationship in themselves.

Comments/questions #36:

Line 302 - The authors apparently meant Figure S4 here. Please correct this and also check figures numbering throughout the manuscript.

Response:

Thanks for your comment. We have fixed this error.

Comments/questions #37:

Line 357-366 - This paragraph is very poorly written. Please carefully rewrite it and show clearer what message it conveys.

Response:

Thanks for your comment. We have fixed this error.

Comments/questions #38:

Figures S2 and S3 have only titles but no details. Please provide details after

the titles.

Response:

Thanks for your comment. We have fixed this error.

Comments/questions #39:

Figure S4E - Indicate in the figure which integration approach is on the left and which is on the right.

Response:

Thanks for your comment. We have fixed this error.

Comments/questions #40:

Tables S1 and S2 do not have titles. Please provide the titles.

Response:

Thanks for your comment. We have fixed this error.

Comments/questions #41:

I have unpacked the rar archives with the Table S3 parts but could not open them. "The files is corrupted" message popped up when I tried opening the files. Please check the files.

Response:

Thanks for your comment. There may have been a file corruption issue caused by oversized files. New file has been uploaded (address: [ZixinQin/Dataset-for-Cross-platform-transcriptomic-data at Update \(github.com\)](https://github.com/ZixinQin/Dataset-for-Cross-platform-transcriptomic-data)).

April 24, 2023

Prof. Wei-Hua Chen
Huazhong University of Science and Technology
Department of Bioinformatics and Systems Biology, College of Life Science and Technology
Luoyu Road 1037
Wuhan 430074
China

Re: Spectrum05369-22R1 (Cross-platform transcriptomic data integration, profiling and mining in *Vibrio cholerae*)

Dear Prof. Wei-Hua Chen:

Your manuscript has been accepted, and I am forwarding it to the ASM Journals Department for publication. You will be notified when your proofs are ready to be viewed.

Sincerely,

Beile Gao
Editor, Microbiology Spectrum
